# Hypersensitivity of Intrinsically Photosensitive Retinal Ganglion Cells in Migraine Induces Cortical Spreading Depression

**DOI:** 10.3390/ijms25147980

**Published:** 2024-07-22

**Authors:** Eiichiro Nagata, Motoharu Takao, Haruki Toriumi, Mari Suzuki, Natsuko Fujii, Saori Kohara, Akio Tsuda, Taira Nakayama, Ayana Kadokura, Manaka Hadano

**Affiliations:** 1Department of Neurology, Tokai University School of Medicine, Isehara 259-1193, Japan; n_fujii@tokai.ac.jp (N.F.); s-kohara@tokai.ac.jp (S.K.); taira_nakayama@hotmail.com (T.N.); cardcoola@gmail.com (A.K.); 2bmm1115@gmail.com (M.H.); 2Department of Human and Information Science, Tokai University, Hiratsuka 259-1292, Japan; takao@tokai.ac.jp; 3Department of Acupuncture and Moxibustion, Shonan Keiiku Hospital, Fujisawa 252-0816, Japan; h.toriumi@ymail.ne.jp (H.T.); smw_suzuki@yahoo.co.jp (M.S.); 4Bioresearch Center Co., Ltd., Tokyo 101-0032, Japan; tsuda.phantom@gmail.com

**Keywords:** migraine, photophobia, intrinsically photosensitive retinal ganglion cell (ipRGC), cortical spreading depression (CSD)

## Abstract

Migraine is a complex disorder characterized by episodes of moderate-to-severe, often unilateral headaches and generally accompanied by nausea, vomiting, and increased sensitivity to light (photophobia), sound (phonophobia), and smell (hyperosmia). Photophobia is considered the most bothersome symptom of migraine attacks. Although the underlying mechanism remains unclear, the intrinsically photosensitive retinal ganglion cells (ipRGCs) are considered to be involved in photophobia associated with migraine. In this study, we investigated the association between the sensitivity of ipRGCs and migraines and cortical spreading depression (CSD), which may trigger migraine attacks. The pupillary responses closely associated with the function of ipRGCs in patients with migraine who were irradiated with lights were evaluated. Blue (486 nm) light irradiation elicited a response from ipRGCs; however, red light (560 nm) had no such effect. Melanopsin, a photosensitive protein, phototransduces in ipRGCs following blue light stimulation. Hypersensitivity of ipRGCs was observed in patients with migraine. CSD was more easily induced with blue light than with incandescent light using a mouse CSD model. Moreover, CSD was suppressed, even in the presence of blue light, after injecting opsinamide, a melanopsin inhibitor. The hypersensitivity of ipRGCs in patients with migraine may induce CSD, resulting in migraine attacks.

## 1. Introduction

Headaches have recently attracted attention as a global health issue. The number of patients with migraines worldwide in 2019 was approximately 581.76 million, representing an increase of 16% since 1990 [1,2]. In Japan, migraine affects 6.0–8.6% of the population [3,4,5], impacting individuals and the economy [6]. Migraine is a complex disorder characterized by moderate-to-severe headaches accompanied by concomitant symptoms, such as photophobia, phonophobia, and hyperosmolarity. Photophobia, defined as an abnormal sensitivity to light, is one of the most frequently reported accompanying symptoms among patients with migraine [7]. However, it has also been reported by patients with dry eyes, corneal neuropathy, blepharospasm, and traumatic brain injuries [8]. Patients do not generally present with photophobia as the chief complaint; instead, they may report sensitivity to light in situations wherein most individuals do not experience sensitivity. Some patients, particularly those with migraine, recognize this sensitivity to light. Therefore, wearing sunglasses in a bright environment is beneficial for patients with migraine and those with light sensitivity. Moreover, patients with migraine prefer low-intensity, low-color-temperature lighting [9].

Green light-emitting diodes (LEDs) reduce the number of headache days suffered by patients with migraine. However, blue LEDs (480 nm) elicit the highest amount of discomfort from glare for patients with migraine [10]. Consequently, migraine attacks occur. Intrinsically photosensitive retinal ganglion cells (ipRGCs) are sensitive to blue light with a 480 nm wavelength. The discovery of ipRGCs that signal the intensity of light on the retina has sparked discussions regarding their potential role in the pathophysiology of migraine. ipRGCs recognize light and dark and provide non-imaging vision [11,12]. ipRGCs have different spectral sensitivity characteristics compared to cones and rods (λmaxs: ipRGC 480 nm (humans and mice); cones 430, 531, and 561 nm (humans); cones 360 and 508 nm (mice); rods 498 nm (humans and mice)) [13]. Moreover, ipRGCs project to the olivary pretectal nucleus and are significantly associated with pupillary reflexes [14]. In particular, the miosis process occurring after removing a visual stimulus reflects the light response characteristics of ipRGCs and is most sensitive to a wavelength of around 480 nm [15,16,17].

Melanopsin, an opsin (OPN4), has been detected in ipRGCs [18]. It is a G-protein-coupled receptor-binding molecule that depolarizes photoreceptors, discharges electrical spikes, and innervates several areas of the brain, thereby influencing physiology, behavior, perception, and mood [19].

The discovery of photophobia in blind patients with migraine has emphasized the key role of ipRGCs in photophobia. Notably, patients with migraine who become blind owing to the degeneration of rods and cones continue to experience photophobia [20]. Therefore, in this study, we investigated the relationship between the pathophysiology of migraine and the function of ipRGC by focusing on the pupillary response.

Cortical spreading depression (CSD) has been hypothesized to be the underlying mechanism of migraine with aura [21]. Substantial evidence that indirectly supports this hypothesis has been gathered from studies that used animal models. Furthermore, studies reporting the prevalence of CSD among humans with brain injury have clearly demonstrated that this phenomenon can occur in the human brain [22]. However, the role of CSD in migraine remains uncertain, and key questions regarding the initiation of this event remain unanswered.

The pupillary reflex is a noninvasive indicator of the photoreactivity of ipRGCs in humans [14,15] and can be useful in examining the increased photoreactivity of ipRGCs in patients with migraines. Meanwhile, the increased activity of ipRGCs in mice is thought to lower the threshold of CSD, raising susceptibility to migraine attacks.

Therefore, in the present study, we aimed to investigate the relationship between ipRGCs and CSD and whether the sensitivity of ipRGCs triggers migraine attacks. Elucidating the mechanism of this pathophysiology may help to prevent migraine attacks.

## 2. Results

### 2.1. Hyperreactivity of ipRGCs in Patients with Migraine Compared with that in Healthy Controls

No significant differences in pupil diameter changes were observed between the migraine patients and healthy controls under a red LED light. The degree of miosis of the pupil was similar in both groups; however, irradiation with a blue LED resulted in stronger miosis than irradiation with a red LED.

Notably, the speed of miosis in patients with migraine was significantly lower than that in healthy controls after discontinuing irradiation with a blue LED light. These findings suggest that ipRGCs in the eyes of patients with migraine are more sensitive to irradiation with a blue LED light than those in the eyes of healthy controls (Figure 1 and Figure 2a,b; Appendix A).

Subsequently, the interval between the discontinuation of irradiation with light and the pupil diameter returning to baseline was examined to determine the pupillary attenuation rate. The attenuation rates of patients with migraine were significantly higher than those of the controls after irradiation with blue LED light in both the first (controls: 27.8 ± 7.2; migraines: 36.3 ± 9.7, *p* = 0.038) and second halves (controls: 5.8 ± 5.4; migraines: 14.4 ± 10.9, *p* = 0.031), while there was no significant difference between the first (controls: 28.9 ± 14.0; migraines: 18.1 ± 28.8, *p* = 0.286) and second halves (controls: 8.0 ± 10.6; migraines: −1.7 ± 22.9, *p* = 0.225) after irradiation with red LED light.

The attenuation rate was the integrated value of the pupil contraction rate at baseline (when no light stimulation was induced) divided by the integrated value of the contraction rate upon light stimulation.

These findings suggest that the ipRGCs in the eyes of patients with migraine were more sensitive than those in the eyes of controls.

### 2.2. Sensitivity of ipRGCs Affects CSD

As shown in Appendix A, CSD was induced in mice in a dark room using KCl at a concentration of 0.225 M. The cerebral blood flow (CBF) increased gradually after CSD. CSD was induced with KCl at a concentration of 0.2 M under irradiation with incandescent light. The DC potential and the regional cerebral blood flow (rCBF) returned to baseline, and the left eye of the mouse under analysis was irradiated with blue LED light after 5 min. CSD was induced with KCl at a concentration of 0.175 M. Blue LED light was shone after the rest period, and opsinamide, a melanopsin inhibitor, was injected peritoneally after 15 min. CSD was induced with KCl at a concentration of 0.2 M (Figure 3). These experiments were performed in triplicate. The findings revealed that although irradiation with blue LED light and the blocking of melanopsin resulted in no significant differences, stimulating ipRGCs may induce CSD (Figure 4).

## 3. Discussion

The participants in this experiment were sighted persons, and the mice were wild-type and had normal retinas. In addition to ipRGCs, cones and rods were also thought to produce photoreactions. However, the maximum spectral wavelength and spectral sensitivity curves of cones and rods differ from those of ipRGCs. We postulate that the λmax of the blue LED used in this experiment (464 nm for humans, 473 nm for mice) most strongly stimulates ipRGCs rather than photoreceptor cells, such as cones and rods.

The findings of this study suggest that the ipRGCs in the eyes of patients with migraine are hypersensitive in comparison with those in the eyes of the controls. This photosensitivity during migraines may be due to the hypersensitivity of ipRGCs. In addition, animal experiments on CSD revealed that bright light, especially blue-LED-light irradiation (to which ipRGCs are hypersensitive), is more likely to induce CSD than a dark environment. The threshold for the development of CSD may be reduced under irradiation with blue LED light.

CSD, a slowly propagating wave of depolarization that is followed by suppression of brain activity, is a remarkably complex event. It involves dramatic changes in neural and vascular functions. Since its discovery in the 1940s, CSD has been hypothesized to be the mechanism underlying migraine with aura. This phenomenon may also occur to some extent in migraine without aura [22].

Tissue susceptibility and suprathreshold triggers affect CSD. Potassium facilitates the induction of CSD, and its clearance depends on glial cells. Lauritzen [23,24] reported that the lowest glial-to-neuronal cell ratio can be observed in the primary visual cortex in humans. Thus, CSD is more likely to be initiated occipitally, as K+ clearance is less effective in the occipital cortex. CSD propagation occurs more effectively in the posterior–anterior direction than in the opposite direction in mice.

A recent report linked CSD to the onset of migraine attacks [22]. The neural pannexin 1 (Panx 1) megachannel is opened in neuronal cells when CSD is induced, and caspase 1 is activated. High-mobility group box 1 (HMGB1) is subsequently released from the neurons, which activates the nuclear factor κB in astrocytes. CSD-induced neuronal megachannel opening may promote sustained activation of trigeminal afferents via parenchymal inflammatory cascades that reach the glia limitans [25].

Some studies delineated a novel neural pathway that can conduct photic signals from the retina to the trigeminovascular neurons in the thalamus [26,27,28]. The neuronal activity of the nociceptive pathway underlying migraine pain is modulated at the level of the posterior thalamus via direct input received from ipRGCs. This is evident from the findings that light enhances the activity of thalamic trigeminovascular neurons in a manner that resembles the light-based activation of melanopsinergic RGCs, which constitute a subset of dura-sensitive thalamic neurons located mainly in the posterior-most region of the thalamus. Melanopsinergic RGCs receive monosynaptic input from RGCs and the axons of dura-sensitive thalamic neurons, whose activity is enhanced by light. They subsequently project into multiple cortical areas, including primary and secondary somatosensory, motor, retrosplenial, parietal association, and visual cortices. In addition, some dura-sensitive thalamic neurons, especially those located in the posterior and lateral posterior thalamic nuclei, project into the primary and secondary visual cortices. CSD activates the direct descending modulation of the thalamus and brainstem (the central pathway) [22]. The thalamus transforms CSD into pain. Moreover, it is the point at which ipRGCs become more sensitive to light stimulation. CSD exacerbates photosensitivity. However, the opposite pathway was also feasible in the present study. Thus, photosensitivity facilitates the induction of CSD through the thalamus, leading to headaches. This finding suggests that ipRGCs are involved in the development and regulation of CSD. Trauma, cerebrovascular accidents, and neuropeptides such as CGRP are known triggers of CSD [29]. The hypersensitivity of the melanopsin induced by melanopsin-mediated transduction in ipRGCs influences photosensitivity and circadian rhythms.

CSD is caused by a variety of triggers, and it is possible that CSD can also be caused by hypersensitivity to sound (phonophobia) or smell (osmophobia), which are common sensitivities in migraine. However, the answer remains unclear because cells like ipRGCs that are active during photosensitivity have not been discovered. In addition, whether the response of ipRGCs differs by gender may be related to hormone dynamics, but at present, there does not appear to be any gender-related response.

ipRGCs release melanopsin upon light stimulation. Recently, Higuchi et al. [30] reported that the I394T mutation in the melanopsin gene enhances the photopupillary reflex in humans. This mutation was detected more often in patients with migraines than in the controls, suggesting that ipRGC photoreactivity may have been enhanced (unpublished data). In addition to photosensitivity, melanopsin is also involved in circadian rhythms. Light-responsive intracellular calcium release in ipRGCs leads to the release of melanopsin, which may increase the likelihood of inducing CSD.

A limitation of this study is its small sample size; nevertheless, the trends can be understood clearly. Although these findings do not provide evidence that melanop

sin causes CSD, it is clear that melanopsin plays a role in inducing CSD.

## 4. Materials and Methods

### 4.1. Analysis of Function of ipRGCs Associated with Pupillary Light Responses in Patients with Migraine and Controls

This study was approved by the Ethical Committee of Tokai University (18R093). Informed consent was obtained from all patients and controls for participation and for publication of identifying information/images in an online open-access publication. All methods were performed in accordance with the relevant guidelines and regulations.

Ten patients with migraine (two and eight patients with migraine with and without aura, respectively [three men and seven women]; mean age, 37.4 ± 8.1 years) were interviewed and enrolled in this study. Two headache specialists diagnosed the participants with migraine based on the International Classification of Headache Disorders, Third Edition, criteria, (Appendix A) [31]. Nine age-matched healthy controls without migraine were also enrolled.

An electronic pupilometer (ET-200, NEWOPTO Inc., Kanagawa, Japan) was used to assess the changes in pupillary diameter. This device can automatically measure the changes in pupil diameter in the right eye when the left eye is irradiated with a red or blue LED light. The participants were instructed to stay in a dark room for 15 min to facilitate dark adaptation. The electronic pupilometer was worn similar to special goggles such that it could measure the pupil diameter and attenuation rate of the pupil diameter in the right pupil when the left pupil was irradiated by LED light (Figure 1). Due to the consensual pupillary reflex, stimulating only one eye with light also constricts the pupil of the other eye. Thus, in this experiment, it was possible to optically stimulate only the left eye and measure the pupillary reflex of the right eye. In addition, the pupillary reflex could be accurately measured while avoiding optical artifacts caused by the irradiation of light onto the retina. The left eye was irradiated with a red LED light (λmax, 632 nm; half-band width, 10 nm; 65.54 cd; light intensity, 15.88 log photons/cm^2^/s) for 20 s 15 min after wearing the goggles. Irradiation with red LED light was subsequently discontinued, and the change in pupil dilation was measured for 35 s. The same procedure was repeated using blue LED light (λmax, 464 nm; half-band width, 23 nm; 69.29 cd; light intensity, 15.88 log photons/cm^2^/s) after a 5 min interval. The data were analyzed to determine the attenuation rate of the pupil diameter (Figure 1). This protocol was repeated twice.

### 4.2. Analysis of Changes in CSD Response to Light in Mice

All experiments involving animals were performed after receiving approval (No. 224007) from the Animal Ethics Committee of Tokai University. All experimental procedures were performed in accordance with the guidelines of the University for the care and use of laboratory animals. This study is reported in accordance with ARRIVE guidelines (https://arriveguidelines.org, accessed on 3 July 2024). The CSD experiment was conducted by modifying the protocols described in previous studies [32,33]. In brief, 10 male C57BL/6 J mice (body weight: approximately 20–25 g) were anesthetized with isoflurane (2.0–2.5% in room-condition air at a flow rate of 250 mL/min). As reported in a previous study, CSD levels in female mice change depending on the fluctuation of various hormones, such as estrogen; thus, we conducted these experiments on male mice only [34]. The head of each mouse was fixed to a head holder modified to be flexible around the horizontal axis (SG-4N; Narishige Scientific Inc. Laboratory, Tokyo, Japan). The mice were intubated intratracheally to provide artificial ventilation 65–70 times/min at a rate of 15 mL/kg using a small-animal ventilator (Harvard Apparatus, USA). Body temperature was maintained at around 37 °C using a heating pad and thermos controller (ThermoStar, Bioresearch Center Co., Ltd., Tokyo, Japan).

Figure 5 presents a schematic representation of the surgery. In brief, two holes were drilled into the right hemisphere of the skull, and the dura mater was removed. A hole of approximately 1 mm in diameter centered at the coordinates 5 mm posterior and 2 mm lateral to the bregma was created in the posterior portion to facilitate KCl stimulation. A hole of <1 mm diameter centered 2 mm lateral and 2 mm caudal to the bregma was created in the parietal bone to facilitate the placement of the recording electrode. A Ag/AgCl DC electrode (tip diameter = 200 μm; EEG-5002Ag; Bioresearch Center Co., Ltd.) was placed such that it made contact with the dura underneath the parietal hole. It was fixed using dental cement afterwards. A DC (direct current) potential was applied at 1–100 Hz and digitized at 1 kHz using a differential head stage and differential extracellular amplifier (Model 4002 and EX1; Dagan Co., Minneapolis, MN, USA). The probe (BF52; Advance Co., Ltd., Tokyo, Japan) of a laser Doppler flowmeter (LDF; ALF 21, Advance Co., Ltd.) was positioned on the intact skull at a point near the parietal hole centered 4 mm lateral and 2 mm posterior to the bregma to monitor the rCBF. We also inserted a silver–silver electrode fiber into the upper limb muscles of the mice and used it as a reference electrode. A multichannel recorder (PowerLab 8/30; AD Instruments, Ltd., Sydney, Australia) was used to store continuous recordings of DC potentials and rCBF. LabChart software (LabChart (Japanese) 8.1, AD Instruments, Ltd.) was used for offline analysis. CSD was induced via chemical stimulation of the dura using various concentrations of KCl solutions (0.1–0.3 mol/L, 5 μL) after confirming that all parameters were stable for at least 10 min. The induction of CSD was confirmed based on the presence of a distinct DC potential defect, typical fluctuations in rCBF, and propagation to the distal portion (Figure 2). CSD induction was performed thrice after washing the cortical surface with normal saline to prevent the residual KCl solution from acting on the nociceptors in the surrounding tissue.

The light source (incandescent, 14.68 log photons/cm^2^/s; blue LED, 13.77 log photons/cm^2^/s) was placed 3 cm ahead of the left eyes of the mice, and tropicamide phenylephrine hydrochloride, a mydriatic drug, was used to dilate the pupils.

### 4.3. Evaluation of the CSD Threshold

Appendix A illustrates the time schedule of CSD. The mice were placed in a dark room for 15 min initially, and the concentration of KCI required to induce CSD was recorded. The same experiment was conducted under the condition of irradiation with incandescent or blue LED light. Opsinamide, an inhibitor of melanopsin, was injected intraperitoneally under the condition of irradiation with blue LED light.

### 4.4. Data Analysis

A multichannel recorder (PowerLab 8/30; AD Instruments Pty. Ltd., Bioresearch Center Co., Ltd.) was used to continuously record and store the rCBF and DC potentials. The data were analyzed offline using LabChart software (AD Instruments Pty. Ltd., Bioresearch Center Co., Ltd.). The time courses of rCBF and DC potential responses were determined by considering the time point at which the DC potential at the proximal portion began to decrease to zero. The troughs and peaks of the rCBF and DC potential were estimated. The period until the recovery rate reached a minimal value was defined as the duration of CSD. The response time and delay time of each parameter were determined for each CSD episode and averaged for each application of KCl.

Student’s *t*-test was used to perform statistical comparisons between the two groups. Average data are presented as means ± standard error of the mean (SEM), and a *P*-value of 0.05 was considered statistically significant.

Statistical analysis was conducted with repeated one-way analysis of variance (ANOVA) for three lighting conditions (incandescent light, blue light, and blue light + opsinamide) to evaluate lighting and pharmacological effects on thresholds for CSD. Post-hoc tests were performed with repeated one-way ANOVA followed by Bonferroni’s multiple comparisons. When sphericity was rejected, Hyunh–Feldt correction was used. Statistical significance was set at *p* < 0.05. The data are shown as means ± SEM. Data were analyzed using SPSS statistical software package (IBM SPSS ver. 22 for windows, Tokyo, Japan).

Data analysis for humans and mice was performed using the *t*-test and ANOVA (SPSS 29, IBM, Tokyo, Japan).

## 5. Conclusions

Hypersensitivity of ipRGCs was observed in patients with migraine. CSD was more easily induced with blue light than with incandescent light using a mouse CSD model. Moreover, CSD was suppressed, even in the presence of blue light, after injecting opsinamide, a melanopsin inhibitor. The hypersensitivity of ipRGCs in patients with migraine may induce CSD, resulting in migraine attacks.

## Figures and Tables

**Figure 1 ijms-25-07980-f001:**
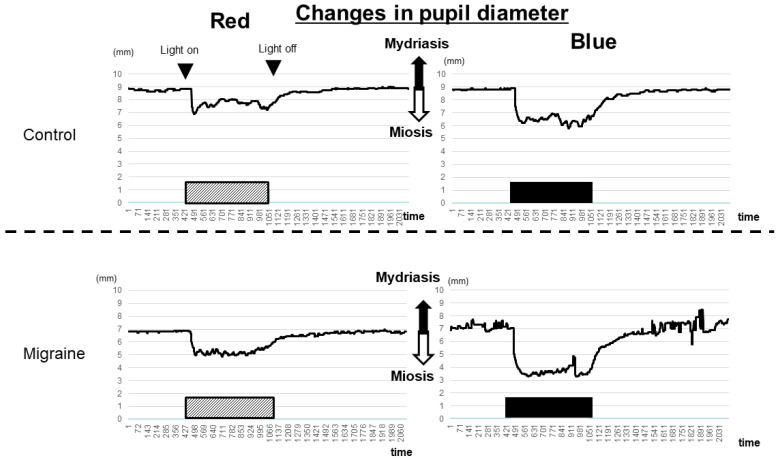
The representative alterations in pupil diameters in patients with migraine and controls. The upper panels show the alterations in pupil diameter in the controls, whereas the lower panels show the alterations in pupil diameter in the patients with migraine. Irradiation with light induced miosis (downward direction), whereas discontinuation of irradiation resulted in mydriasis (upper direction). When irradiation with blue-LED light was discontinued, the time required for patients with migraine to return to their baseline pupil diameter was longer than that in controls. Diagonal boxes: red-LED light irradiation time; black boxes: blue-LED light irradiation time; LED: light-emitting diode.

**Figure 2 ijms-25-07980-f002:**
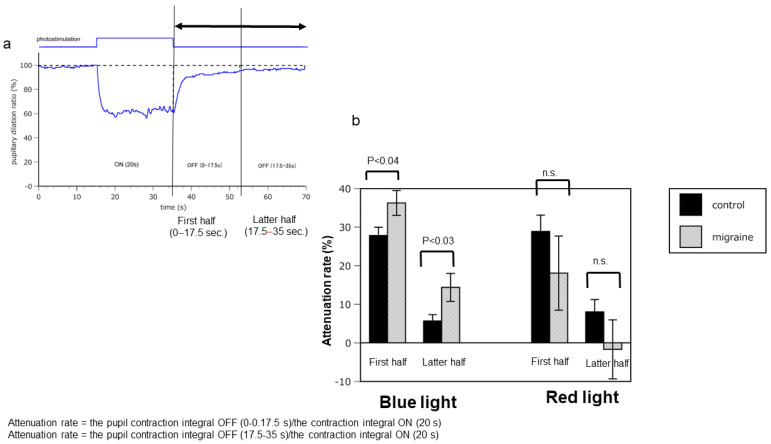
The attenuation rate of pupils in patients with migraine and controls. (**a**) The changes in pupil diameter. The pupil constricts when irradiated with light (miosis: down direction). The pupil dilates when irradiation is discontinued (mydriasis: upper direction). The time required for the pupil diameter to return to baseline after pupil miosis was divided into two parts: the first half and the second half. (**b**) Under blue-LED light irradiation, the attenuation rate was significantly increased during both the first and second halves in the patients with migraine compared with that in controls. However, no significant difference was observed between the two groups under red-LED irradiation. Attenuation rate = pupil contraction integral OFF (0–0.17.5 s)/contraction integral ON (20 s). LED: light-emitting diode.

**Figure 3 ijms-25-07980-f003:**
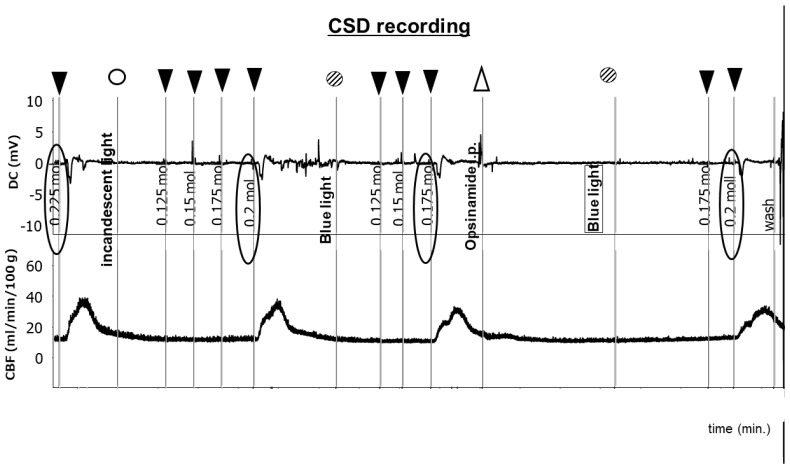
Representative recording of CSD induction in mice. The upper graph shows the changes in direct current (DC) potential, whereas the lower graph shows the changes in regional cerebral blood flow (rCBF). Black inverted triangle, KCl addition; white triangle, opsinamide intraperitoneal injection; white circle, incandescent light is on; diagonal circle, blue LED light is on. The DC potential decreased when cortical spreading depression (CSD) was induced; at the same time, CBF increased. CSD was induced using 0.225 M KCl after 15 min in a dark room. Under incandescent light, CSD was induced using 0.2 M KCl. Under blue LED light, CSD was induced using 0.175 M KCl. Moreover, when opsinamide was injected intraperitoneally into the mice, the CSD threshold increased under blue LED light. LED: light-emitting diode.

**Figure 4 ijms-25-07980-f004:**
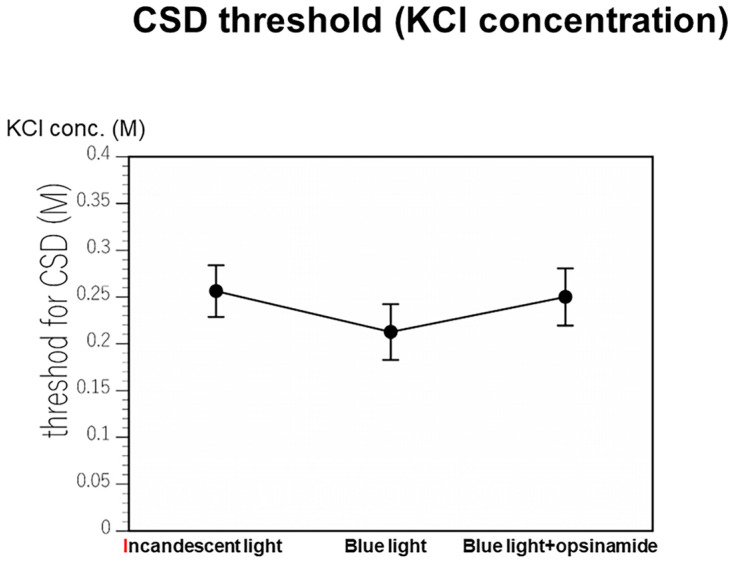
Summary of the CSD threshold related to KCl concentrations. The cortical spreading depression (CSD) threshold under blue LED light was lower than that under incandescent light irradiation. The addition of opsinamide under these conditions increased the CSD threshold. LED: light-emitting diode.

**Figure 5 ijms-25-07980-f005:**
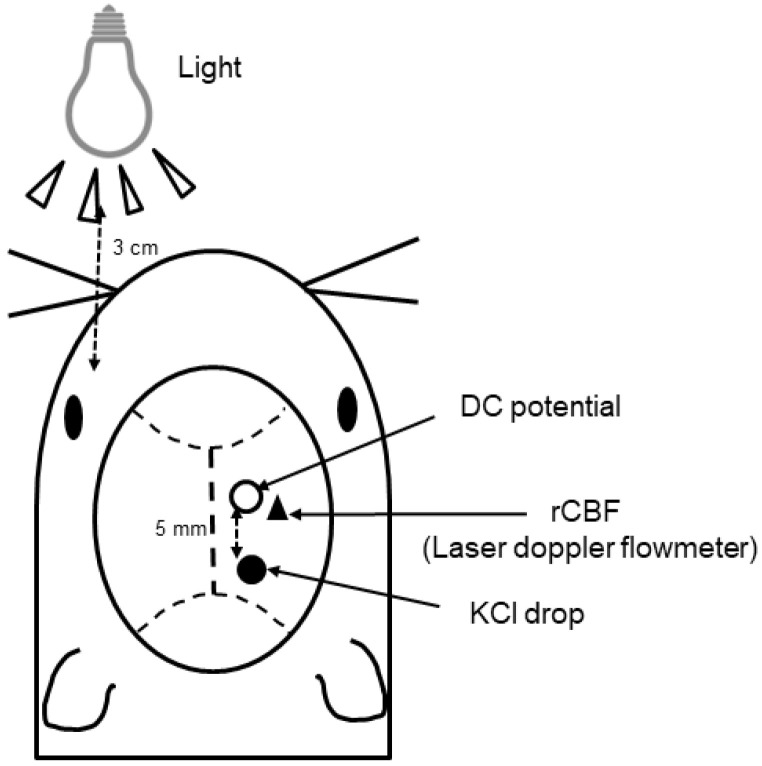
A diagram depicting the experimental setting for inducing CSD. CSD, cortical spreading depression; rCBF: regional cerebral blood flow.

## Data Availability

The datasets generated and/or analyzed during the current study are available from the corresponding author on reasonable request.

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
