# Peer review of "Hypersensitivity of Intrinsically Photosensitive Retinal Ganglion Cells in Migraine Induces Cortical Spreading Depression"

_ijms, 2024, doi:10.3390/ijms25147980_

Round 1

Reviewer 1 Report

Comments and Suggestions for Authors

Q1: LED light?

Should be “a” LED light

Q2:line 88-90:No significant differences were observed between the patients with migraine and  healthy controls in terms of the changes in pupil diameter induced by irradiation with red LED light.

No significant differences in pupil diameter changes were observed between migraine patients and healthy controls under a red LED light. 

Q3 :line 164: The photosensitivity observed during migraine may be attributed to this finding.

This sentence is not clear.

I think that you may rephrase your sentence such as This photosensitivity during migraines may be due to the hypersensitivity of ipRGCs?

Q4: Introduction and discussion: very clear and well-organized. 

Author Response

Q1: LED light?

Should be “a” LED light

[Response] According to the reviewer’s suggestion, we changed “LED light” to “a LED light” throughout our manuscript.

Q2:line 88-90:No significant differences were observed between the patients with migraine and  healthy controls in terms of the changes in pupil diameter induced by irradiation with red LED light.

No significant differences in pupil diameter changes were observed between migraine patients and healthy controls under a red LED light. 

[Response] According to the reviewer’s suggestion, we revised the part of our manuscript.

Q3 :line 164: The photosensitivity observed during migraine may be attributed to this finding.

This sentence is not clear.

I think that you may rephrase your sentence such as This photosensitivity during migraines may be due to the hypersensitivity of ipRGCs?

 [Response] According to the reviewer mentioned, we revised this part of sentences.

Q4: Introduction and discussion: very clear and well-organized. 

[Response] Thank you very much.

Reviewer 2 Report

Comments and Suggestions for Authors

This is a valuable study designed to address a part of the underlying mechanism of migraine. The intrinsically photosensitive retinal ganglion cells (ipRGCs) are proposed to be involved in photophobia. Here, the authors investigated the association between the sensitivity of ipRGCs and cortical spreading depression (CSD) in migraine. The pupillary responses were evaluated for the function of ipRGCs in patients with migraine irradiated with lights. Blue light elicited a response from ipRGCs; however, red light had no effect. Hypersensitivity of ipRGCs 25 was observed in patients with migraine. CSD was more easily induced with blue light using a mouse CSD model. CSD was suppressed by injecting opsinamide, a melanopsin inhibitor. The authors collectively concluded that hypersensitivity of ipRGCs in patients with migraine may induce CSD, resulting in migraine attacks. The authors have indicated one limitation of this study, as the small sample size. What other limitations are here? sources of bias and if the study can be generalized or not in terms of the findings. What are the confounding factors in this study? both in the human and animal studies? Can the authors elaborate further about the CSD that can be a combination of several factors at the same time, and not separated factors such as light? One can assume that smell, and sound can also trigger the CSD for instance. Please add whether this can be a sex-related response as migraine is predominant in women. 

Author Response

This is a valuable study designed to address a part of the underlying mechanism of migraine. The intrinsically photosensitive retinal ganglion cells (ipRGCs) are proposed to be involved in photophobia. Here, the authors investigated the association between the sensitivity of ipRGCs and cortical spreading depression (CSD) in migraine. The pupillary responses were evaluated for the function of ipRGCs in patients with migraine irradiated with lights. Blue light elicited a response from ipRGCs; however, red light had no effect. Hypersensitivity of ipRGCs 25 was observed in patients with migraine. CSD was more easily induced with blue light using a mouse CSD model. CSD was suppressed by injecting opsinamide, a melanopsin inhibitor. The authors collectively concluded that hypersensitivity of ipRGCs in patients with migraine may induce CSD, resulting in migraine attacks. The authors have indicated one limitation of this study, as the small sample size. What other limitations are here? sources of bias and if the study can be generalized or not in terms of the findings. What are the confounding factors in this study? both in the human and animal studies? Can the authors elaborate further about the CSD that can be a combination of several factors at the same time, and not separated factors such as light? One can assume that smell, and sound can also trigger the CSD for instance. Please add whether this can be a sex-related response as migraine is predominant in women. 

Thank you for your nice comments on our manuscript. I’d like to answer the questions that the reviewer asked.

The authors have indicated one limitation of this study, as the small sample size. What other limitations are here?

[Response] The function of ipRGCs can only be indirectly seen by pupillary responses. In addition, because it is impossible to measure the amount of melanopsin contained in ipRGCs, their responses can only be inferred by using melanopsin inhibitor, opsinamide.

What are the confounding factors in this study?

[Response] We performed a series of experiment in a dim room, therefore, a weak room light may stimulate ipRGCs. However, intensities of our light sources were far stronger than room light intensity. Light intensity threshold of an ipRGC is reported to be much higher than cone and rod photoreceptors (Takao et al., 2017). It appears that intensity of weak room light is not enough to depolarize ipRGCs1).

1) Takao, M., Fukuda Y., Morita, T. (2017) A novel intrinsic electroretinogram response in isolated mouse retina. Neuroscience 357, 262-371.

both in the human and animal studies?

[Response] This research is a study of a phenomenon common to humans and animals.

Can the authors elaborate further about the CSD that can be a combination of several factors at the same time, and not separated factors such as light?

One can assume that smell, and sound can also trigger the CSD for instance

Please add whether this can be a sex-related response as migraine is predominant in women. 

[Response] According to the reviewer’s suggestion, the following sentences were added in page 6, line 209-214.

“CSD is caused by a variety of triggers, and it’s possible that CSD can also be caused by hypersensitivity to sound (phonophobia) or smell (osmophobia), which are common sensitivities in migraine. However, it’s unclear because cells like ipRGCs that are active during photosensitivity have not been discovered. In addition, whether the response of ipRGCs differs by gender may be related to hormone dynamics, but at present there does not appear to be any gender-related response.”

Reviewer 3 Report

Comments and Suggestions for Authors

This manuscript included 2 experiments. A human study used 10 patients with migraine and 9 healthy controls. Various conditions may affect the results, including the frequency and severity of migraine attacks, accompanying medication status, and visual characteristics including visual acuity. Photosensitivity in these patients may be various. Readers have much interest in the association between photosensitivity and pupillary responses. 

Author Response

This manuscript included 2 experiments. A human study used 10 patients with migraine and 9 healthy controls. Various conditions may affect the results, including the frequency and severity of migraine attacks, accompanying medication status, and visual characteristics including visual acuity. Photosensitivity in these patients may be various. Readers have much interest in the association between photosensitivity and pupillary responses. 

[Response] Thank you very much for the reviewer’s comment.